# Extracellular Vesicles: New Players in the Mechanisms of Sepsis- and COVID-19-Related Thromboinflammation

**DOI:** 10.3390/ijms24031920

**Published:** 2023-01-18

**Authors:** Martina Schiavello, Barbara Vizio, Ornella Bosco, Emanuele Pivetta, Filippo Mariano, Giuseppe Montrucchio, Enrico Lupia

**Affiliations:** Department of Medical Science, University of Turin, 10126 Turin, Italy

**Keywords:** extracellular vesicles, sepsis, COVID-19, thromboinflammation

## Abstract

Sepsis and COVID-19 patients often manifest an imbalance in inflammation and coagulation, a complex pathological mechanism also named thromboinflammation, which strongly affects patient prognosis. Extracellular vesicles (EVs) are nanoparticles released by cells into extracellular space that have a relevant role in cell-to-cell communication. Recently, EVs have been shown to act as important players in a variety of pathologies, including cancer and cardiovascular disease. The biological properties of EVs in the mechanisms of thromboinflammation during sepsis and COVID-19 are still only partially known. Herein, we summarize the current experimental evidence on the role of EVs in thromboinflammation, both in bacterial sepsis and in COVID-19. A better understanding of EV involvement in these processes could be useful in describing novel diagnostic and therapeutic applications of EVs in these diseases.

## 1. Introduction

Sepsis is defined as a life-threatening organ dysfunction due to a dysregulated host response to infection [1]. It is the final common pathway to death from most infectious diseases worldwide, including bacterial and viral infections such as Severe Acute Respiratory Syndrome Coronavirus 2 (SARS-CoV-2) [2]. The World Health Organization has listed sepsis as one of the global health priorities over the next few years, [3] and this is the most common cause of emergency admission to intensive care units (ICUs) in Europe [4,5].

Evidence has shown that the relationship between inflammation and coagulation, described by the term “thromboinflammation”, is critical in the pathogenesis of sepsis [6].

The same pathogenic mechanism has also been indicated as the most crucial, leading to elevated morbidity and mortality in patients affected by Coronavirus Disease-2019 (COVID-19), the complex clinical syndrome caused by the recently emerged Severe Acute Respiratory Syndrome Coronavirus 2 (SARS-CoV-2) [7,8,9], which can lead to acute respiratory distress syndrome, sepsis, and multi-organ failure [10].

Although the precise mechanisms are not fully understood, inflammation through complement activation and cytokine release, platelet hyperactivity and apoptosis (thrombocytopathy), as well as coagulation abnormalities (coagulopathy) play critical roles in this complex scenario [10,11].

Extracellular vesicles (EVs) are membrane-derived vesicles released into extracellular space by all cell types, and they emerged as novel mediators of cell-to-cell and organ-to-organ crosstalk in many physiological and pathological conditions [12,13].

The diversity of EVs lies in different mechanisms of biogenesis, as well as in their different cellular origin [12]. EVs can be found in many biological fluids, such as plasma [14], saliva [15] and urine [16]. In addition, their ability to modulate inflammation has suggested EVs as potential novel biomarkers and next-generation biological therapeutics. Finally, there is growing interest in their ability to exchange functional components between cells. EVs may, indeed, communicate with cells and affect their phenotype or function via surface molecule-triggered uptake and intracellular signalling via cargo content [13].

The biological potential of EVs in the mechanism of sepsis- and COVID-19-related thromboinflammation is still only partially known; however, the rapid development of novel molecular platforms is contributing to envisage and develop theranostic applications of EVs in critical illness also.

The aim of this review is to outline the involvement of EVs in thromboinflammation associated with sepsis and COVID-19. Based on the well-established pathogenic role of thromboinflammation in the development of organ damage in critically ill patients affected by these diseases, we will discuss the current experimental evidence on the role of EVs in thromboinflammation, both in bacterial sepsis and in COVID-19, aiming to explore potential targets in the coagulation processes useful to translate the knowledge into the discovery of novel potential diagnostic and therapeutic values of EVs in these diseases.

## 2. Thromboinflammation in Sepsis and COVID-19: Differences and Overlaps

### 2.1. Thromboinflammation or Immunothrombosis?

In 2004, Tanguay and colleagues were the first to use the term “*thromboinflammation*” to indicate the platelet–leukocyte interaction mediated by P-selectin and P-selectin glycoprotein ligand 1 (PSGL-1) implicated in stent restenosis [17]. Inflammation-associated thrombosis, now known as thromboinflammation, occurs commonly in a broad range of human disorders.

Thrombosis is well defined as an exaggerated hemostatic response, leading to the formation of an occlusive blood clot obstructing blood flow through the circulatory system. By comparison, inflammation is the term applied to the complex protective immune response to harmful stimuli, such as pathogens or damaged cells. Increasingly well-defined is the recognition that inflammation stimulates thrombosis, and in turn, thrombosis promotes inflammation [18].

In 2013, Engelmann and Massberg coined the term “immunothrombosis” to explain a tricky and mutual interaction, whereby, on the one hand, the activation of coagulation cascade triggers the immune system, cooperating with the identification, containment and destruction of pathogens [19], whereas, on the other hand, the innate immune cells promote the development of thrombi [20]. Today, thromboinflammation or immunothrombosis are considered interchangeable terms indicating dysregulation of the physiologic anti-thrombotic and anti-inflammatory functions of endothelial cells, which negatively influences hemostasis and favors thrombus deposition both in micro and macro-vasculature [18,21].

### 2.2. Cellular Mechanisms of Sepsis-Related Thrombosis

Sepsis is a life-threatening systemic illness associated with the invasion of the bloodstream by pathogens such as bacteria, viruses and fungi [1]. It is considered an extremely complex illness both on the cellular and on the molecular level, involving, among different pathophysiologic mechanisms, an imbalance in the inflammatory response, immune dysfunction, mitochondrial damage, coagulopathy, and immune network abnormalities, ultimately leading to multi-organ failure (MOF) and death [22,23].

It is known that septic patients are at high risk of developing thrombotic complications ranging from widespread microvascular involvement, such as disseminated intravascular coagulation (DIC), to venous thromboembolism, arising as deep vein thrombosis (DVT) or pulmonary embolism (PE). The development of these potentially fatal complications is believed to be triggered by a common final cascade of events characterized by serious injury to the microvasculature of affected tissues and organs and by the excessive activation of the coagulation system resulting in increased thrombus formation [6,24,25].

At the beginning of the infectious process, components of the bacterial cell wall such as pathogen-associated molecular patterns (PAMPs) are recognized by pattern recognition receptors (PRRs) present on the surface of endothelial cells, platelets, and leukocytes [26]. PRRs transduce signals leading to the release of inflammatory chemokines and cytokines, and of other inflammatory mediators that increase the expression of leukocyte-adhesion molecules [27]. As a consequence, the natural anticoagulant system on endothelial cells is altered and tissue factor (TF) production by monocytes and endothelial cells is increased. The relevance of TF in promoting thromboinflammation in sepsis has been confirmed using pharmacological TF inhibitors or through the inhibition of TF expression in mice exposed to endotoxin, which leads to diminished coagulation and inflammation and improved survival [28].

The location and grade of thromboinflammatory lesions are largely determined by the severity of damage to the endothelial cells (ECs) lining vessels. Usually, endothelium expresses both anti-inflammatory and anti-thrombotic substances, which antagonize leukocyte adhesion to and platelet accumulation [29]. Upon vascular injury, platelets are recruited to the site of damage. Their activation and adherence to the vessel wall lead to the release of platelet agonists, such as adenosine diphosphate (ADP), which further induce paracrine platelet activation and platelet aggregation [30].

Another crucial point is the activation of neutrophils that release nuclear DNA and granular protein, known as neutrophil extracellular traps (NETs) [31], with a strong antibacterial activity. Furthermore, platelets attached on the surface of neutrophils increase the formation of NETs that damage microcirculation, promote immunothrombosis, and lead to diffuse intravascular coagulation, since they facilitate the formation of thrombus acting as a scaffold [32].

### 2.3. COVID-19-Related Thromboinflammation

SARS-CoV-2 is an enveloped, single-stranded RNA virus [33] that uses the angiotensin-converting enzyme 2 (ACE2) receptor for internalization in the host cells, aided by transmembrane serine protease 2 (TMPRSS2) receptor [34]. ACE2 receptor is highly expressed in many human cells, including nasal and oral epithelial cells, endothelial cells, and myocardial cells [35].

SARS-CoV-2 infection may cause diffuse lung alveolar and endothelial cell damage with severe inflammation and increased vascular permeability leading to acute respiratory distress syndrome (ARDS). In addition to systemic inflammation and severe respiratory disease, COVID-19 is frequently complicated by the development of a hypercoagulable state and by the appearance of thrombotic events, which represent a primary cause of mortality in these patients [7,8,11,36,37,38]. Histopathological findings, indeed, have reported pulmonary microthrombi in 57% of SARS-CoV-2 infection patients, compared with 24% of H1N1 influenza patients, a subtype of influenza A virus [39], with increased angiogenesis and pulmonary microthrombosis, respectively, three and nine times more prevalent in COVID-19 patients [40].

Although the appearance of a pro-thrombotic profile is now a well-recognized hallmark of COVID-19 [36,40,41,42], the mechanisms underlying the development of thromboinflammation are not completely elucidated yet.

In the process of SARS-CoV-2 infection progressing to a systemic and severe disease, multiple proinflammatory cytokines, which include interleukin (IL)-1, IL-6, tumor necrosis factor (TNF)-α, and chemokines, recruit more innate immune cells (such as macrophages, neutrophils and dendritic cells) to produce additional inflammatory cytokines, in a loop known as cytokine storm. This increases vascular dysfunction and thrombosis and favors the development of multiorgan failure [43,44]. Vascular dysfunction is primarily sustained by the abundant expression on endothelial cells of the ACE2 receptor, which leads to increased virus tropism in blood vessels [45]. In general, the vascular endothelium is considered the first responder of the host defense. Once homeostasis is disrupted by SARS-CoV-2 infection, endothelial cells lose their anti-thrombotic capacity as a consequence of glycocalyx damage [44]. In addition, on their surface they express procoagulants and proapoptotic factors, such as TF and phosphatidylserine (PS), thereby leading to the exposure of the basement membrane and the activation of the coagulation cascade [46,47]. SARS-CoV-2 also directly activates platelets and exacerbates the thromboinflammatory cascade by promoting platelet–neutrophil binding, which, in turn, increases the formation of NETs, activates additional inflammatory responses, and increases other prothrombotic pathways [48,49].

In this complex cellular and molecular interplay, our research group has recently demonstrated that thrombopoietin (THPO), a humoral growth factor involved in the proliferation and differentiation of megakaryocytes in the bone marrow, appears as an additional crucial candidate mediator of platelet hyper-activation and immunothrombosis/thromboinflammation in COVID-19 [50].

### 2.4. Do Sepsis and COVID-19 Overlap?

The complex inter-relationship and similarities between sepsis and COVID-19 are topics that have recently emerged with particular emphasis [51,52,53].

Data obtained in hospitalized COVID-19 patients has revealed that serum cytokine and chemokine levels are high in these patients, at levels comparable to those found in patients with sepsis [54,55].

Some researchers have also pointed out that severe and critically ill COVID-19 patients meet the diagnostic criteria for sepsis and septic shock according to the Sepsis-3 International Consensus, and, thus, recommend using the term ‘viral sepsis’ instead of the terms severe or critical illness because it is more appropriate [56,57].

Patil and co-authors have suggested that SARS-CoV-2 itself likely causes sepsis as a consequence of various mechanisms, which include immune-dysregulation, respiratory dysfunction leading to hypoxemia, and metabolic acidosis due to circulatory dysfunction [58].

Multiorgan failure seen in COVID-19 could also be explained by hypoxia and circulatory disorders that occur as a consequence of microvascular dysfunction, analogously to what is described in sepsis [59].

Finally, other authors have suggested that microvascular dysfunction may also contribute to hypoxia and subsequent organ failure by interrupting the blood flow to the lungs due to disseminated intravascular coagulation and micro-embolism [60].

## 3. Extracellular Vesicles

### 3.1. Definition and Classification

EVs are a heterogeneous population of membrane-bound particles released by cells that contain both distinct cargoes of proteins, lipids, metabolites, nucleic acids, which include small non-coding RNA such as microRNAs, long non-coding RNAs (lncRNAs), and circular RNAs (circRNAs), and organelles that reflect the parental cells and modulate the functions of recipient cells [12].

EVs are released from all cell types, including those circulating in the blood such as platelets, monocytes, and granulocytes [61,62].

A univocal classification of EVs is still lacking, and, thus, continuously evolving. With the aim of answering this need, in 2014, the International Society for Extracellular Vesicles (ISEV) enacted the Minimal Information for Studies of Extracellular Vesicles (MISEV) guidelines [63], which were then updated in 2018 [64].

The current ISEV guidelines settle that “EVs” remain a collective term describing a complex continuum of vesicles of different sizes and compositions, resulting from various mechanisms of formation and release [63,64].

Conventionally, EVs are classified into different subclasses based on size and biogenesis. The studies of the scientific community mostly focused on two classes, namely medium/large EVs or microvesicles (MVs) (>200 nm), which are produced through the outward budding of the cell membrane, and small EVs or exosomes (<200 nm), produced from the endosomal pathway and formed through inward invagination of the endosomal membrane to form the multi-vesicular body [12,64,65].

Analogously to the use of cell-specific surface markers for cell characterization, cluster of differentiation (CD) markers are widely used to characterize different EV subpopulations.

Small EVs are enriched with non-tissue specific markers, such as tetraspanins (CD63, CD9 and CD81), proteins involved in multi-vesicular body formation (ALIX and TSG101), membrane trafficking proteins (RAB proteins and annexins), and MHC class I (HLA-A/B/C) [64]. On the contrary, large vesicles, such as MVs, carry surface markers that derive from the composition of the cellular plasma membrane upon release, and also express cell/tissue-specific markers, such as CD45 (if derived from immune cells), CD41 and CD42a (from platelets), CD14 (from monocytes), or other cell-specific surface molecules [66].

### 3.2. Methods to Study Extracellular Vesicles

Although ISEV has provided technical protocols and recommendations for EV isolation [64,67], a standardized procedure has not yet been established [68], especially for biofluids, such as blood, where viscosity, as well as fat and protein content, are highly variable. These factors may affect EV purity and yield and require the isolation protocols to be adjusted according to the biofluid of interest [69,70]. A standardization of pre-analytical steps is crucial in order to minimize the artefacts in EV analysis from biological fluids. On the contrary, the use of cell culture medium enables a more controlled environment for EV isolation [71].

Therefore, the choice of the EV isolation method largely depends on the sample analyzed, and on planned downstream analysis and applications.

Several researchers have compared different methodologies to improve the efficiency of the isolation and characterization of EVs [61,69,70,72,73,74]. The need for robust, standardized and reproducible EV-isolation methods is an essential requirement, especially for the translation in a clinical setting [75].

Some authors sustain that the combination of different methods is often the best option for EV isolation since the different origin, complex nature, and heterogeneity of EVs require sophisticated isolation approaches [73].

The main isolation techniques include ultracentrifugation (UC), immunoaffinity capture (IC), size-exclusion chromatography (SEC), ultrafiltration (UF) and precipitation [64,72,76]. Each one of these methods offers some advantages and disadvantages compared to the others. The most popular isolation method still in use is based on UC, which is surely cost-efficient and widely accepted, but also time-expensive and burdened with the undesired co-purification of non-EV-associated proteins [61]. Immunoaffinity capture methods are usually based on the coating with specific CD antibodies and theoretically allow only one EV subpopulation to be specifically isolated [74]. Size-exclusion chromatography, ultrafiltration and precipitation are relatively widely used methods [61]. In particular, SEC has been used as a method to enrich EVs whilst depleting protein contaminants, and is often used in EV-based omics analysis [77,78].

For more details on the discussion of the adequate isolation method of EVs, we address the readers to several comprehensive reviews or position statements already published on this topic [61,79,80].

After isolation, EV populations need to be characterized for intended downstream applications, for which there is a variety of techniques available. These methods include dynamic light scattering/nanoparticle tracking analysis (NTA), which is the most popular quantitative method for EV particle analysis [81], flow cytometry, the most widely employed technique to study large extracellular vesicles [82], and transmission electron microscopy (TEM), a routine method that has been used with great success for the study of EVs [83].

### 3.3. EVs in Cell-to-Cell Communication

EVs are pivotal elements in cell-to-cell communication, in both physiological and pathological states, via the transfer of selective biomolecular cargos to recipient cells in an autocrine, paracrine, or endocrine manner to regulate cell function [84].

Although the uptake of EVs by different cell types is a well-described phenomenon, the processes directing EV entry into recipient cells remain poorly understood.

The initial meeting with the recipient cell is based on a variety of receptors and/or molecules located on the plasma membrane. This binding is recognized and drives intercellular signaling events, finally establishing the process of cell entry, as well as other downstream effects [85].

An attractive paradigm of EV-mediated cell–cell communication depends on the existence of an “address system” where the EV membrane composition provides a high degree of selectivity in terms of targeting specific recipient cell types [12].

Various evidence in vivo sustains that EVs with a distinct composition are able to target specific organs to elicit microenvironment changes in a remarkably selective way [86].

Due to their stability in the circulation, low immunogenicity, biocompatibility and biodegradation, in addition to their ability to deliver proteins, lipids, and nucleotides from one cell to another, and specifically to selected cell types, EVs have started to attract attention for their potential applications as therapeutic delivery systems in various medical fields [87], including immunology [88], cancer research [89], cardiovascular [90], inflammatory diseases [91], and infection [92].

Considering the preliminary nature of the evidence on therapeutic applications of EVs in sepsis, in the following paragraphs, we specifically focus on highlighting the role of EVs as contributors to the pathological mechanisms described above.

## 4. EV Role in Sepsis-Related Thromboinflammation

In sepsis, increased circulating levels of proinflammatory and procoagulant EVs are well documented and may contribute to coagulation disorders likely involved in the pathophysiology of DIC [62,93,94,95,96].

EVs may exhibit direct procoagulant properties, in part due to PS expressed on their surfaces, a cell membrane phospholipid that supports the assembly of coagulation enzymes and TF [97]. Septic patients have increase circulating PS-positive EVs that are mainly released by endothelial cells, monocytes and platelets [98,99,100].

Zhang et al. [100] reported that endothelial cells treated with EV-containing serum obtained from septic patients exhibited more exposed PS than those treated with serum from healthy controls. These data support the concept that procoagulant properties of effector cells can be transmitted to target cells by EVs.

Tripisciano et al. [101] investigated the thrombogenicity of platelet-derived EVs and compared the contributions of PS and TF exposure on thrombin generation. They demonstrated that annexin V (which binds and masks PS), not anti-TF antibodies, efficiently inhibits this effect, indicating that thrombin generation is primarily due to the exposure of PS on platelet-derived EVs [101].

In addition to PS exposure, EVs can carry TF, whose activity contributes to their pro-coagulant activity in various diseases, including cancer, acute coronary syndromes, and sepsis itself [102,103,104]. Higher amounts of TF-positive EVs are associated with an increased occurrence of DIC in sepsis [99]. However, the role of TF-positive EVs in septic thromboinflammation is still considered controversial. For more detailed information on this specific topic, the readers can refer to some interesting reviews [105,106].

New data support the notion that EVs from different cell types have distinct characteristics and play different pathogenic roles in thromboinflammation during sepsis.

Platelet-derived EVs are considered dominant in DIC, but those derived from other cell types, including leukocytes and endothelial cells, are believed to also contribute to procoagulant mechanisms in sepsis [105,107,108,109,110,111].

Platelet-derived EVs adhere to platelets, leukocytes, and endothelial cells and induce pro-inflammatory and pro-thrombotic functions in these cells [107,112,113] since they contain a distinctive subset of proteins and microRNAs involved in hemostasis and thrombosis [107]. Recently, it was seen that platelet-derived exosomes promote excessive NET formation in sepsis, through modulating the Akt/mTOR-related autophagy pathway partly by the transfer of selected microRNAs [108]. This suggests that exosomes released from activated platelets are essential mechanisms in sepsis-induced thromboinflammation [107,108,112,113].

Leukocyte-derived EVs, originating from neutrophils, monocytes/macrophages and lymphocytes, may also contribute to the disruption of vascular homeostasis via their cytoplasmatic contents, for instance through reactive oxygen species (ROS). Mortaza et al. have indeed shown that EVs isolated from septic rats reproduce hemodynamic, inflammatory and oxidative stress patterns of sepsis in healthy rats [109]. These effects were mediated by increased superoxide anion production, NF-kB activation, NO synthase (NOS)-2 expression, and NO overproduction [109]. Similarly, Mastronardi et al. found that EVs isolated from septic patients induce tissue-selective expression of pro-inflammatory proteins in rats, whereas EVs derived from healthy subjects do not [110]. These results further support the possibility of distant dissemination of EVs and their involvement in the pathogenesis of MOF in sepsis.

Other evidence suggests that activated ECs increase their procoagulant activity during sepsis by enhancing the production of EVs that bind to neutrophils. Endothelial-derived EVs stimulate the oxidative activity with increased binding to leukocytes in patients with severe systemic inflammatory response syndrome [111].

Profiling exosome cargos, including proteins, mRNAs, lncRNAs and miRNAs, is crucial in order to identify molecular markers of sepsis-related thromboinflammation, and, thus, elucidate the functions and the related regulatory mechanisms of EVs in this process, potentially leading to the development of new tailored therapeutic approaches in sepsis. Only one experimental study until now has addressed this topic using a multi-omics approach, showing that serum exosomes from septic mice may exert a therapeutic effect through cytokine storm suppression, inhibition of complement and coagulation system activation, endothelial cell junction stabilization, and vascular permeability reduction [114]. Further studies on the regulation of gene expression mediated by miRNAs contained in EVs and on their relation to the thromboinflammatory mechanisms present in sepsis would certainly be needed.

Finally, recent evidence suggests that EV-mediated transfer of mitochondrial content may alter metabolic and inflammatory responses [115], similar to what is described in neuroinflammation [116]. Understanding the potential role of mitochondrial in sepsis-related thromboinflammation and deepening this aspect associated with EVs could also reveal yet unexplored mechanisms.

The main concepts of these paragraphs are schematically depicted in Figure 1.

## 5. EV Role in COVID-19-Related Thromboinflammation

In support of the role of EVs in promoting thrombosis in COVID-19, a systematic analysis of surface antigen expression of EVs circulating in COVID-19 patients showed 37 antigens involved in inflammation, platelet activation, coagulation processes, and endothelial dysfunction [117]. The cytokine storm occurring during COVID-19 might influence the cell release of substantial amounts of procoagulant EVs that may act as clotting initiation agents. In particular, the elevated levels of TNF-α in COVID-19 patients’ serum strongly correlate with TF (CD142) expression onto the surface of EVs [117], as well as with its procoagulant activity [118]. Several studies [119,120,121,122] have reported a significant increase in circulating TF-expressing EVs that correlated with D-dimer levels, von Willebrand Factor, circulating leukocytes, and inflammatory markers, assuming the contribution of TF-EVs in disease determining severity and thrombosis in COVID-19 patients. However, the identification of the exact cellular origin of TF-exposing EVs, as well as of the underlying pathogenic mechanism, remains challenging and requires additional study.

Controversial data have been reported on PS exposure from EVs isolated from COVID-19 patients: some studies have reported an increase [122,123], and others a decrease [124]. These differences may be related to the heterogeneity of patient cohorts.

Furthermore, EVs from COVID-19 patients, in particular large EVs, are able to directly induce a significantly higher thrombin generation compared to EVs isolated from healthy donors [125].

EVs are released by activated platelets, leukocytes, and endothelial cells [126,127]. In many cardiovascular diseases, EVs have been shown to be able to propagate inflammation and coagulation, contributing to tissue injury and thrombosis [128]. These mechanisms are likely contributing to exacerbate vascular conditions also in COVID-19 [129]. For instance, platelet-derived EVs were shown to promote the formation of NETs during SARS-CoV-2 infection through Toll-like receptor (TLR)-2-dependent mechanisms [130]. Moreover, neutrophils-derived EVs synergize with NETs in neutrophil activation and endothelial adhesion through a high-mobility group box protein 1 (HMGB1)/TLR pathway [131].

Proteomic analysis of EVs has been used in different pathological conditions, including sepsis [132] and cancer [133], with the aim of elucidating those metabolic pathways that are modulated by the action of EVs. This same methodologic approach has shown that circulating EVs in plasma from COVID-19 patients exhibit a distinct procoagulant profile. Especially in the most severe patients, this is reflected by a profound proteomic change, mainly related to coagulation activation, complement cascade, and platelet degranulation [134]. Moreover, multi-omics analysis of EVs isolated from hospitalized COVID-19 patients, comprising metabolome, lipidome and proteome studies, allows the description of a specific signature that correlates with their coagulopathic potential [125].

In addition, microRNA cargo of EVs could contribute to the pathogenesis of thrombotic complications in COVID-19 patients by downregulating two specific miRNAs (miR-145 and miR-885) that promote higher levels of TF and von Willebrand Factor, thus determining a prothrombotic state [135].

The accumulated evidence on the pathophysiological role of EVs in COVID-19-related-thromboinflammation has enhanced our understanding of the relationship between viruses and coagulation and may help formulate effective targeted strategies.

The main concepts of this paragraph are schematically depicted in Figure 1.

## 6. Challenges and Future Directions

The emerging interest in the role of EVs in critical illness is supported by their early involvement in relevant pathobiological functions of vital organs exposed to severe stressors. Although EVs offer innovative theragnostic possibilities in detecting, monitoring, and modulating the onset of tissue dysfunction before the development of overt organ failure, technical challenges based on isolation, characterization methods, and standardization of clinically suitable EV collections represent a significant issue.

The development of EV-based biomarkers is promising, particularly for diseases such as cancer and neurological conditions [136,137].

In recent years, EVs have gained progressive importance in sepsis and, more recently, in COVID-19, two similar but heterogenous critical diseases.

EVs represent novel players involved in the mechanisms of thromboinflammation in these diseases since they concur to promote a switch toward a procoagulant phenotype, as already observed in cardiovascular disorders [90] and cancer [89]. However, a complete picture of EV active functions in sepsis and COVID-19 is still lacking. Therefore, a better understanding of the role of EVs in thromboinflammation is needed, in order to improve our knowledge about the mechanisms underlying these diseases, to be subsequently translated into clinical practice, for effective and personalized management of septic and COVID-19 patients. Further investigations are required in these critical illnesses, which will also take advantage of the next-generation approaches using novel imaging techniques, omics-based methodologies, and organ-on-chip systems to reveal new features of EVs. There is still a significant number of questions to be answered in order to better clarify the involvement of EVs in this scenario. The more one discovers, the more questions arise.

## Figures and Tables

**Figure 1 ijms-24-01920-f001:**
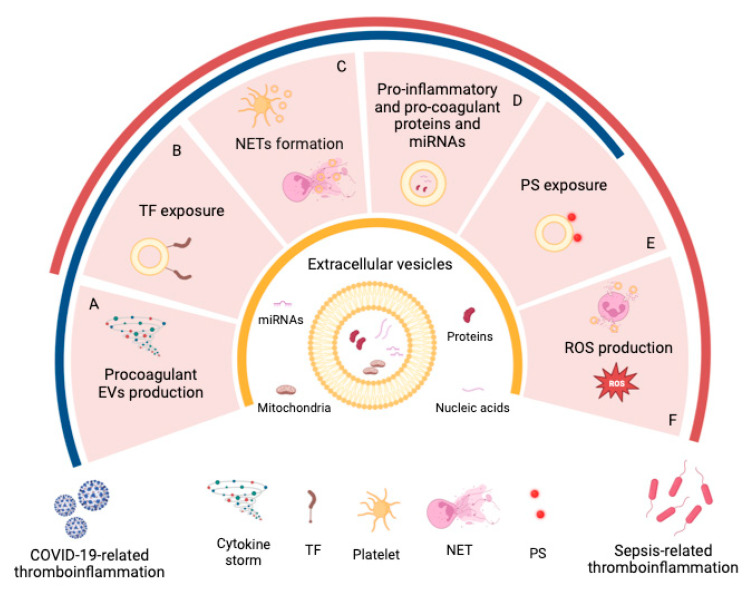
Extracellular vesicles as new players in sepsis- and COVID-19- thromboinflammation. Schematic summary of some of the main roles of EVs in sepsis- (red line) and COVID-19- (blue line) related-thromboinflammation. (**A**) The cytokine storm occurring during sepsis and COVID-19 might influence the cell release of substantial amounts of procoagulant EVs involved in thromboinflammation development. (**B**) Higher amounts of TF-positive EVs are associated with an increased occurrence of both DIC in sepsis and thrombosis in COVID-19. (**C**) Activated platelets release EVs that promote excessive NET formation both in sepsis and COVID-19. (**D**) Protein and miRNA cargo in EVs could contribute to the pathogenesis of thrombotic complications in sepsis and COVID-19 patients. (**E**) EVs exhibit direct procoagulant properties due to PS expressed on their surfaces in sepsis and (probably) in COVID-19 patients. (**F**) Leukocyte-derived EVs contribute to thromboinflammation via excessive ROS production. TF, tissue factor; NETs, neutrophil extracellular traps; PS, phosphatidylserine; ROS, reactive oxygen species. Created with BioRender.com.

## Data Availability

Not applicable.

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
