# Peer review of "Extracellular Vesicles: New Players in the Mechanisms of Sepsis- and COVID-19-Related Thromboinflammation"

_ijms, 2023, doi:10.3390/ijms24031920_

Round 1

Reviewer 1 Report

Dear Authors,

There are few minor suggestions and corrections.

Recent study by Setua et al (PMCID: PMC9780627) has studied the role of COVID19 EVs on Thrombin generation. It is relevant to the topic discussed here.

line 245: address

line 293: masks 

Author Response

We want to thank reviewer 1 for carefully reading our review and for the minor revisions.

Point 1: Recent study by Setua et al (PMCID: PMC9780627) has studied the role of COVID19 EVs on Thrombin generation. It is relevant to the topic discussed here.

Response 1: We agree with the suggestion of Review 1 and have discussed the study in lines 361-363; 378-380.

Point 2: line 245: address; line 293: masks 

Response 2: We have made the indicated correction in line 242 and we have replaced the word “marks” with “masks” in line 290.

Reviewer 2 Report

The manuscript is an interesting review about the role of EVs in thromboinflammation.  It is well written, organized and the concepts are clearly explained.

Minor points:

1.       Line 103. I think the word “cytokine” should be plural “cytokines”

2.       Line 228. I think the references 46-48 are not about comparisons of methodologies for EVs analysis. Here, authors can cite references 61, 70-82.

3.       Lines 268-269. It seems to me that it should be “evidenceS” (in plural) and “changeS” (in plural).

4.       Lines 356 and 381. The abbreviation “vWF” is not defined in the text and probably it is not necessary. In both lines the complete name can be written: von Willebrand Factor .

After addressing these points, in my opinion, the manuscript can be accepted for publication.

Author Response

We want to thank reviewer 2 for carefully reading our review and for the minor revisions.

Point 1: Line 103. I think the word “cytokine” should be plural “cytokines”.

Response 1: We corrected it in line 101.

Point 2: Line 228. I think the references 46-48 are not about comparisons of methodologies for EVs analysis. Here, authors can cite references 61, 70-82.

Response 2: We have replaced with the appropriate references in line 225.

Point 3: Lines 268-269. It seems to me that it should be “evidenceS” (in plural) and “changeS” (in plural).

Response 3: We corrected it in lines 265-266.

Point 4: Lines 356 and 381. The abbreviation “vWF” is not defined in the text and probably it is not necessary. In both lines the complete name can be written: von Willebrand Factor.

Response 4: We corrected it in lines 353 and 383.